# Intersectionality-Informed Sex/Gender-Sensitivity in Public Health Monitoring and Reporting (PHMR): A Case Study Assessing Stratification on an “Intersectional Gender-Score”

**DOI:** 10.3390/ijerph20032220

**Published:** 2023-01-26

**Authors:** Emily Mena, Katharina Stahlmann, Klaus Telkmann, Gabriele Bolte

**Affiliations:** 1Department of Social Epidemiology, Institute of Public Health and Nursing Research, Faculty of Human and Health Sciences, University of Bremen, 28359 Bremen, Germany; 2Health Sciences Bremen, University of Bremen, 28359 Bremen, Germany; 3Department of Medical Biometry and Epidemiology, University Medical Center Hamburg-Eppendorf, 20251 Hamburg, Germany

**Keywords:** health monitoring, health reporting, intersectionality, gender, score, mental health, self-rated health, hypertension, decision trees, gradient boosting models

## Abstract

To date, PHMR has often relied on male/female stratification, but rarely considers the complex, intersecting social positions of men and women in describing the prevalence of health and disease. Stratification on an Intersectional Gender-Score (IG-Score), which is based on a variety of social covariables, would allow comparison of the prevalence of individuals who share the same complex intersectional profile (IG-Score). The cross-sectional case study was based on the German Socio-Economic Panel 2017 (n = 23,269 age 18+). After stratification, covariable-balance within the total sample and IG-Score-subgroups was assessed by standardized mean differences. Prevalence of self-rated health, mental distress, depression and hypertension was compared in men and women. In the IG-Score-subgroup with highest proportion of males and lowest probability of falling into the ‘woman’-category, most individuals were in full-time employment. The IG-Score-subgroup with highest proportion of women and highest probability of falling into the ‘woman’-category was characterized by part-time/occasional employment, housewife/-husband, and maternity/parental leave. Gender differences in prevalence of health indicators remained within the male-dominated IG-Score-subgroup, whereas the same prevalence of depression and self-rated health was observed for men and women constituting the female-dominated IG-Score-subgroup. These results might indicate that sex/gender differences of depression and self-rated health could be interpreted against the background of gender associated processes. In summary, the proposed procedure allows comparison of prevalence of health indicators conditional on men and women sharing the same complex intersectional profile.

## 1. Introduction

Applying an intersectionality-informed perspective in quantitative health research can help to recognize the heterogeneity of indicators and processes across different intersections of social positions and promote a better understanding of health and social experiences [1]. The term “intersectionality” was introduced by legal scholar Kimberlé Crenshaw as part of her critique of antidiscrimination doctrine, feminist theory, and antiracist politics. Crenshaw [2] called attention to the intersectionality of social experiences and experiences of discrimination among women of colour in the United States. She illustrated that the intersectional experiences of women of colour as “multiply burdened” women cannot be translated into concrete policy demands when the prevailing frameworks commonly used to address gender *or* racial/ethnic discrimination are single-issue focused. Meanwhile, the concept of intersectionality has evolved to include a variety of categories of differences [3].

The project AdvanceDataAnalysis is part of the collaborative research network AdvanceGender, which aims to promote sex/gender-sensitive and intersectional health research and health reporting [4,5]. In AdvanceDataAnalysis, we focus on the translation of social theory into quantitative methodology. The term sex/gender is used to express the entanglement of the biological concept of sex and the social concept of gender from an embodiment perspective [6]. We seek to identify and develop intersectionality-informed and sex/gender-sensitive strategies for quantitative analyses, which might be adequate for Public Health Monitoring and Reporting (PHMR). The term “intersectionality-informed” used here means that we have not taken a purely intersectional approach but have carried out an intersectional gender analysis as defined by the World Health Organisation [7]. Accordingly, we focus on sex/gender as the anchor point of our analyses and aim to explore its intersections with further sociocultural, sociodemographic, and socioeconomic variables from an intersectionality-informed perspective [4,8,9].

The negative impact of discrimination experiences on mental and physical health is well documented [10]. Integrating an intersectional perspective into our understanding of the relationship between experiences of discrimination and health requires an understanding of how discrimination can be better captured when different intersectional profiles of populations are considered to analyse health disparities [11,12]. A first step in examining discrimination processes in the context of population health might be to first take a descriptive, intersectionality-informed approach. In this way, existing differences in people’s health status can be examined in relation to different social positions [3]. Describing differences in the prevalence of health indicators from an intersectionality-informed perspective has traditionally been based on cross-classification [1,4]. Because cross-classification depends on a large number of strata, a sufficient sample size of the resulting subgroups cannot always be guaranteed, and the interpretability of results, such as the prevalence of health indicators, might be limited with an increasing number of intersectionality-informed variables considered for analysis. In PHMR, a prevalence comparison that describes the health status of men and women with the same intersectional profile could support the identification and description of health inequalities in an even more precise and sex/gender-sensitive manner than is possible with the usually applied stratification by the male/female binary. In addition, it could help address the statistical limitations of complex, cross-classified descriptive analyses of the prevalence of health indicators and allow for the consideration of a greater variety of categories of difference, thereby strengthening intersectionality-informed PHMR and its sensitivity to sex/gender.

Consistent with our project goal described above, we applied stratification on a predicted probability score, which we termed Intersectional Gender-Score (IG-Score). The estimated IG-Score is the probability of “being a woman” conditional on the included intersectional covariables (hereafter referred to as intersectional variables). After dividing men and women with similar IG-Scores into equally sized subgroups, men and women within each of the IG-Score subgroups have approximately similar IG-Score values (intersectional profiles), because they were assigned to a subgroup, based on the covariables used to calculate the IG-Score according to their probability of being a woman. If men and women were completely equal, the distributions of men’s and women’s IG-Scores would completely overlap. In contrast, a lack of overlap between the distributions of men’s and women’s IG scores would indicate the presence of social forces that influence the distribution of power and resources by promoting sex/gender stereotypes, because the intersectional variables used to calculate the IG-Score are essentially interchangeable in case of complete gender equality between men and women. Accordingly, the IG-Score approach quantifies constructed “feminization” (high probability of being a woman based on the intersectional variables included) as the lack of overlap between the IG-Score distribution of men and women: less gender equality corresponds to higher IG score discrepancy.

The underlying assumption of our case study is that men’s and women’s social experiences may differ significantly with respect to aspects such as participation in the labour force and responsibilities in domestic work [13]. This, in turn, could be interwoven with other categories of difference, such as disability status, age, or migration background, which could be expressed in terms of different intersectional profiles between men and women. We assume that men and women differ substantially in terms of the probability of “being a woman” depending on the intersectional covariables included. If this was true, we would expect the proportion of men and women within each subgroup to vary across subgroups: the first subgroup consists of individuals with the lowest predicted probability of “being a woman” conditional on the included covariables, and thus should have the highest proportion of men in the subgroup. The last subgroup consists of individuals with the highest predicted probability of “being a woman” conditional on the included covariables and is expected to have the highest proportion of women in this subgroup. Consequently, these two subgroups with the greatest sex/gender imbalance would likely represent the most divergent social experiences of men and women with respect to the intersectional factors under consideration. If there is little or no difference between men and women in the prevalence of the health indicators studied within these subgroups, where men and women share the same intersectional profile, this might suggest that these intersectional gender profiles, as defined by the covariables included in the calculation of the IG-Score, could explain the differences observed in the reporting of sex/gender-stratified prevalence within the overall sample.

Using cross-sectional data from a large representative survey (“The German Socio-Economic Panel”), we examine whether differences in prevalence of health indicators between men and women, which are often captured mainly by stratification according to the binary characteristic “male/female,” are also detectable in subgroups in which men and women have similar intersectional profiles. In particular, the two subgroups with the greatest imbalance between the proportions of men and women (the highest and lowest IG-Scores, respectively) are comprehensively examined. In this way, we aim to capture similarities and differences between men’s and women’s social experiences and health and examine the prevalence of corresponding health indicators within subgroups in which men and women share the same intersectional profile based on a variety of sociocultural, sociodemographic, and socioeconomic variables. In the present case study, we take an intersectionality-informed perspective and focus on health indicators that have already been associated with experiences of discrimination and are often reported as different between men and women: self-rated health, depression, frequent mental distress, and hypertension [10,14,15]. The goal of the present analysis strategy is, first, to investigate whether our assumption that men and women differ substantially in the probability of being a woman as a function of the intersectional covariables included proves true; second, if there is evidence supporting our hypothesis, we will explore potentially existing patterns of heterogeneity of health indicators associated with different profiles of intersecting identities; and third, we will evaluate the possibilities and limitations of stratification based on an IG-Score in light of an intersectionality-informed and sex/gender-sensitive PHMR. To calculate the IG score, we compare the results of traditional regression methods with those of Gradient Boosting Models (GBM) [16] as a classification algorithm, which is based on decision trees, to determine which of the two approaches yields better results in terms of classification accuracy. The hypothesis is that the proposed strategy will more accurately capture the complexity of social experiences than the commonly used stratification by the male/female binary.

## 2. Materials and Methods

### 2.1. Study Sample

The present cross-sectional study is based on data collected in 2017 as part of the “German Socio-Economic Panel” (SOEP) [17]. SOEP provides annual longitudinal data from demographic, socioeconomic, behavioural, and attitudinal measures collected through individual- and household-level interviews in Germany [18]. Households are randomly selected in a multi-stage selection process and participants are mainly interviewed face-to-face. Cross-sectional representation is maintained by the integration of enlargement samples to account for loss to follow-up. Further details about design, methods and nationwide representativeness of the study population of the SOEP have been described elsewhere [17]. The final study sample of the present study comprised overall 23,159 participants aged 18+.

### 2.2. Intersectional Gender-Score Variables

#### 2.2.1. Sex/Gender Variable

Interviewers were asked before handing in the questionnaire to enter the sex of the study participant as obtained from the registration office [19]. This binary sex/gender variable (men, woman) was used as the dependent variable to calculate the predicted probability of “being a woman” based on included intersectional covariables.

#### 2.2.2. Intersectional Covariables

Relying on the Progress-Plus Framework [20], we selected 12 variables of SOEP 2017 capturing different intersectional sociocultural, sociodemographic, and socioeconomic dimensions: age [in years: 18–29, 30–39, 40–49, 50–59, 60–69, 70–79, 80+]; employment status [full-time, part-time, some-time, work + education, education, retired, maternity/parental leave, registered as unemployed, housewife/-husband (self-identified; caring for children, relatives and/or doing housework)]; occupational status [blue-collar worker, white-collar worker, civil servant, freelancer, helping family, apprenticeship, else]; education [high, medium, low, else]; partner in household [yes, no]; children <16 years in household [yes, no]; migration background [none (born in Germany), direct (born in another country than Germany), indirect (born in Germany and father or mother with direct migration background)]; household language [German, other]; feeling lonely [often, seldom/never]; disability status (self-report of legal assignment) [yes, no]; urbanity/rurality [urban, rural]; household help [yes regularly, yes sometimes, no].

### 2.3. Health Indicators

Four health indicators were studied: First, feeling sad in the last four weeks (hereafter referred to as frequent mental distress = FMD, as an approximation to the Centers for Disease Control and Prevention (CDC) definition of 14 or more self-reported mentally unhealthy days in the past 30 days [21]) was assessed by the following question: “I will now read off a number of feelings. For each one, please state how often you experienced this feeling in the last four weeks. How often have you felt sad?” [very often, often, occasionally, rarely, very rarely; binarized for analysis into very often/often vs. occasionally/rarely/very rarely]. Second, depression diagnosis (self-reported): Lifetime prevalence [yes vs. no]. Third, self-rated health was assessed by the following question: “How would you describe your current health?” [bad, poor, satisfactory, good, very good; binarized for analysis into bad/poor vs. satisfactory/good/very good]. Fourth hypertension diagnosis (self-reported): lifetime prevalence [yes vs. no] [19].

### 2.4. Statistical Analysis

Analyses were based on data from all study participants of SOEP 2017 aged 18+ (total sample n = 23,269, men: n = 10,754, women: n = 12,515 women). We used Gradient Boosting Models (GBM) [16] for estimating the predicted probabilities of “being a woman” which constitute the IG-Score. This method is a popular classification algorithm based on decision trees. A major advantage of applying tree-based techniques in intersectional research is that no a-priori knowledge on the covariables used to predict the outcome is necessary and even complex interactions between them can be uncovered. The area under the curve of the receiver operating characteristic (AUC-ROC) was 0.809 deeming it an “excellent” classifier for this task according to Mandrekar [22]. In contrast, a logistic regression resulted in an AUC-ROC of 0.7645. In a second step, observations were allocated to subgroups by the resulting IG-Score estimates. Data analysis was performed in R version 4.1.2 using the packages WeightIt [23] for fitting GBMs and cobalt [24] for assessing covariable balance. Imbalance with respect to any of the covariables can be detected by inspection of the standardized mean differences (SMD) [25]. The SMD is the difference in means of an included covariable between men and women divided by their standard deviation. These measures were calculated for the total study sample and each subgroup. To assess proper balancing, we defined values between 0.00–≤0.10 as negligible imbalance, 0.11–≤0.20 as low imbalance [25,26] and >0.20 as unbalanced following the suggestions from Mamdani et al. [27]. Negligible imbalance ensures that the distribution of baseline covariables does not differ systematically between men and women. The optimal number of subgroups was identified by systematically pre-testing between four to seven subgroups and comparing the corresponding balance diagnostics. In particular, the number of subgroups with the least amount of imbalanced covariables (SMD > 0.1) was chosen. Recent work on applying GBM recommends accounting for missing values by using surrogate splits [28]. However, since previous work in this area suggests multiple imputation prior to GBM to account for missing values [29], we also performed a corresponding sensitivity analysis with multiple imputation by chained equations using the mice package [30]. However, the AUC-ROC did not change compared to using surrogate splits. Another reason for refraining from multiple imputation is that imputed values in the intersectional covariables might introduce additional bias to the standardized mean differences. The best balance was found for GBM with surrogate splits and five subgroups (Appendix A). Additionally, the distributions of the IG-Score for men and women were visually inspected by a density plot (Appendix A). A lack of distributional overlap can indicate that combinations of covariable values might exist which guarantee the study participant to be classified as either male or female with certainty. Thus, 110 men and women of the sample who had no overlap in the distribution of the IG-Score, were excluded [31]. This resulted in a final analysis sample with a total size of n = 23,159.

Appendix A shows the balance of covariables between men and women for the total sample and the five IG-Score subgroups. The total sample showed imbalance with regard to categories of employment status (fulltime; part-time; housewife/husband, maternity/parental leave) and occupational status (blue-collar). Low imbalance was found for categories of occupational status (white-collar, freelance), education (middle), partner in household, and feeling lonely. All the remaining categories of the intersectional covariables within the total sample showed no imbalance. Across the five subgroups living with children under the age of 16 years in the female-dominated subgroup exceeded the threshold of 0.2 only marginally (SMD = 0.22). Low imbalance was found in the male-dominated subgroup for categories of occupational status (blue-collar, white-collar) and education (high, low) as well as in the female-dominated subgroup for age group (60–69 years, 70–79 years), employment status (part-time, maternity/parental leave, housewife/-husband), occupational status (blue-collar, else), and education (high, middle). All the remaining categories of the intersectional covariables across the five subgroups showed no imbalance.

Prevalence of the health indicators along with their 95% confidence intervals were computed for the total sample and within each of the two subgroups with highest sex/gender imbalance (male-dominated and female-dominated subgroups), stratified by male/female. Tests for equality of proportions were employed to test for significant differences in prevalence between men and women for binary health indicators (α = 5%) within and across the male-dominated and female-dominated subgroups.

## 3. Results

The total study sample of 23,159 subjects consisted of 46% men (n = 10,724) and 54% women (n = 12,435). Across the five subgroups, the proportion of men varied from 82% in the male-dominated subgroup to 7% in the female-dominated subgroup and the proportion of women from 18% to 93%, respectively. Appendix A shows the complete description of the total sample and the five subgroups with regard to the considered intersectional covariables stratified by men/women. In the total sample the strongest difference in proportions between men and woman of at least 10% was observed for working full-time, part-time, or as a blue-collar. While the male dominated subgroup contained nearly exclusively full-time employed men and women and hardly anyone working part-time or occasionally, the reverse is true for the female-dominated group. With focus on the subgroups two to four (S2–S4) an ascending gradient was observed for not living with a partner in the household and often feeling lonely. The range in difference of proportions across the five subgroups which exceeded 10% was observed for categories of age group, employment status, occupational status, education, living with a partner or children in the household, and feeling lonely. Very strong differences in proportions across all the subgroups were detected in persons being a housewife/-husband or on maternity/parental leave, who were almost exclusively classified within the female-dominated subgroup. Categories of migration background, household language, disability status, urbanity/rurality, and household help did not exceed a range of difference in proportions of 10% across the five subgroups.

With respect to age patterns similarities between the male-dominated subgroup and the female-dominated subgroup, as well as the subgroups S2–S4 were observed. In contrast to the male-dominated subgroup and the female-dominated subgroup, in S2–S4 a higher proportion of persons was aged 18–29 or 60+ and accordingly, noticeably a higher proportion of persons was either in education or in retirement as well as unemployed. Whereas in the male-dominated subgroup and the female-dominated subgroup approximately 83% were distributed across the age groups 30–59 years, this was the case for only approximately 41% across S2–S4. Despite the observed differences in age patterns when comparing the male-dominated subgroup and the female-dominated subgroup vs. S2–S4, strong differences in proportions of partner in household (range across male-dominated/female-dominated subgroup: 81%–90%; range in mean across S2–S4: 38%–69%) and children <16 years in household (range across male-dominated/female-dominated subgroup: 60%–65%; range across S2–S4: 14%–27%) appeared as well. Overall, the male-dominated subgroup and the female-dominated subgroup were quite comparable with respect to most considered intersectional variables, as the majority of the intersectional categories did not exceed a range of difference in proportions of 10%.

As shown in Appendix A and summarised in Table 1, the most prominent differences appeared with respect to employment and occupational status: the male-dominated group is strongly characterised by 91% persons working full-time and 36% as blue-collars. In contrast, the female-dominated group is strongly characterised by 64% persons working part-time, 63% working as white-collars, 16% being housewife/-husband and 6% on maternity/parental leave.

Table 2 displays the prevalence of FMD, depression, self-rated health, and hypertension within the total sample, the male-dominated subgroup, and the female-dominated subgroup. In the complete sample all health indicators under study showed differences in prevalence between men and women.

Table 3 complements Table 2 with focus on differences in prevalence between men and women when comparing the results within the male-dominated and female-dominated subgroups, where men and women share the same intersectional profile, as well as across all the remaining combinations of men and women constituting the male-dominated and female-dominated subgroups: within the group, which we termed “traditional gender group”, we compare men of the male-dominated subgroup with women of the female-dominated subgroup; within the group, which we termed “non-traditional gender group” we compare men of the female-dominated group with women of the male-dominated group. Furthermore, we compare women with women as well as men with men across both subgroups.

With respect to mental health, highest difference in prevalence was observed in the traditional gender group with a difference of 8.04% for FMD (prevalence in men: 5.41%, prevalence in women: 13.45%) and 4.98% for depression (prevalence in men: 3.47%, prevalence in women: 8.45%). Lowest and statistically non-significant difference in prevalence was observed in the non-traditional gender group with a difference in prevalence of 3.18% for FMD (prevalence in men: 9.32%, prevalence in women: 12.50%) and 0.85% for depression (prevalence in men: 8.07%, prevalence in women: 7.22%). Focusing on direct comparisons between men and women, the same pattern for both health indicators was observed, when ranking the groups from highest to lowest difference in prevalence: traditional gender, male-dominated, female-dominated, non-traditional gender. With respect to self-rated health the only statistically significant difference between men and women of 2.43% was found in traditional gender, with a prevalence of 12.05% in men and 14.48% in women, respectively. With regard to hypertension, differences across all the gender groups comparing men and women were statistically significant (*p* = 0.01–*p* < 0.001) with highest difference in prevalence of 6.50% observed in the non-traditional gender group (prevalence in men: 21.43%, prevalence in women: 14.93%). Lowest difference in prevalence was observed in the traditional gender group with a difference in prevalence of 5.89% (prevalence in men: 20.90%, prevalence in women: 15.01%). The same pattern as in FMD and depression, only reversed, was observed when ranking the groups from highest to lowest difference in prevalence: non-traditional gender, female-dominated, male-dominated, traditional gender.

Focusing on differences within men and within women, statistically significant differences were observed only with respect to FMD (*p* = 0.006) and depression in men (*p* < 0.001). With regard to FMD the difference in prevalence was 3.91% (prevalence in men of the male-dominated subgroup: 5.41%, prevalence in men of the female-dominated subgroup: 9.32%). With respect to depression, the difference in prevalence was 4.60% (prevalence in men of the male-dominated subgroup: 3.47%, prevalence in men of the female-dominated subgroup: 8.07%).

## 4. Discussion

Evaluation of the applied strategy and the process of translating the theoretical concepts of intersectionality and gender into quantitative methodology with stratification on an IG-Score is hereafter discussed with respect to the key findings as well as against the background of some strengths and limitations of the present case study. In the following, our summarised reflections are complemented with recommendations for further development of stratification on an IG-Score in light of Public Health Monitoring and Reporting.

First, standardised mean differences and balance of included covariables in the total sample seem to reflect sex/gender differences with respect to sociocultural, sociodemographic, and socioeconomic aspects well. The total sample showed strong imbalances with regard to employment and occupational status. Across the five subgroups, a clear gradient with respect to working full-time, part-time, or occasionally occurred. As expected, decreasing proportions of men and increasing proportions of women across the five subgroups were observed for working full-time. As gender roles evolve most strongly during the core working years where persons build their career and/or family [13] it seems plausible that the male-dominated subgroup and the female-dominated subgroup showed similarities in age distribution (most men and women were aged 30–59), presence of children under the age of 16 years and whether sharing a household with a partner. When interpreting gender around the core working years, the observed differences between men and women seem plausible as well, as persons being a housewife/-husband or on maternity/parental leave were almost exclusively classified in the female-dominated subgroup (93% women). In contrast, persons working full-time were strongly represented in the male-dominated subgroup (82% men). Since stratification on male/female in PHMR is already implemented as a standard procedure [5], a description of similarities and differences of men and women with regard to a variety of sociocultural, sociodemographic, and socioeconomic covariables might enrich PHMR from an intersectionality-informed and sex/gender-sensitive perspective. This applies all the more as in PHMR often only a binary sex/gender variable is available and hence the whole spectrum of gender identities cannot be taken into account.

Second, as a consequence of the aforementioned, rethinking representativeness of population-based studies from an intersectionality-informed and sex/gender-sensitive perspective might be warranted [32]. It would be interesting to explore how intersections of sex/gender with employment and occupational status in the general population are captured by adjusting prevalence or effect estimates with population-weights of age, binary sex/gender, migration background, and education, which are commonly applied in PHMR. In case the applied population weights do not sufficiently correct for the existing patterns in the general population, further adjustments with commonly available census data such as information about working full time or working as a blue-collar might be advisable.

Third, overall, our approach gave plausible insights into patterns of how a study population can be reflected against the background of possibly “gendered” social experiences. However, defining these social experiences and the mere proportion of men or women in a subgroup without consideration of the whole spectrum of gender identities bears the risk of essentialising individuals and neglecting sex/gender diversity beyond the binary [33]. As such, the proposed strategy might serve as an important first step allowing to further develop an intersectionality-informed and sex/gender-sensitive strategy for health reports. In case information about the whole spectrum of gender identities is not available, then capturing the construct of gender within the binary might still give important insights. The discussion of limitations when assumptions are based only on the sex/gender binary should then be elaborated more in-depth.

Fourth, our results showed that differences in prevalence of FMD, depression, and self-rated health between men and women were highest in the traditional gender group and lowest or even non-existent in the non-traditional gender group. In the male-dominated subgroup, where women and men share the same “male-stereotypical” intersectional profile with respect to the included sociocultural, sociodemographic, and socioeconomic covariables, the difference in prevalence of FMD, depression, and self-rated health between men and women was only lowered marginally. In contrast, in the female-dominated subgroup, where men and women share the same “female-stereotypical” intersectional profile, the difference in prevalence of FMD between men and women decreased from 7.62% to 4.13%, due to a higher prevalence in men classified in this subgroup compared to men in other subgroups. With respect to depression and self-rated health the differences between men and women of this subgroup even dissolved, due to higher prevalence in men as well. In a nutshell, male gender, which in the present work was defined by the intersectional profile of men constituting the male-dominated subgroup, might represent only marginally protective social experiences with view on mental health and self-rated health of women. In contrast, female gender, which in the present work was defined by the intersecting profile of women constituting the female-dominated subgroup, seems to mark social experiences in which higher prevalence of FMD, depression, and self-rated health appears for both, men with a “female-stereotypical” intersectional profile and women regardless of their intersectional profile. No difference in prevalence of depression and self-rated health between men and women in the female-dominated subgroup might indicate that sex/gender differences of these health indicators could be interpreted against the background of gender associated processes [34]. Especially in such cases, reporting on sex/gender differences should be carefully considered so as not to reduce individuals’ complex social experiences to a particular category such as “being a woman.” A further exploration of the social experiences as described above might be of value for PHMR when focusing on mental health or self-rated health and gender.

Fifth, compared to our case study with Classification Tree Analysis (CTA) and focus on mental health [8], the results seem to point into the same direction as summarised in the aforementioned paragraph for stratification on an IG-Score: in CTA we found that gender related factors such as social support, burden due to housework, and unpaid care activities were more important for the identification of subgroups with high prevalence of mental distress than considering only the male/female binary. To some extent this relationship between sex/gender, gender-related factors, and mental health might be reflected by the results of the IG-Score analysis as well, since persons on maternity/parental leave or being a housewife/-husband were exclusively classified in the female-dominated subgroup: highest differences in prevalence of FMD and depression between men and women was found in the traditional-gender group and lowest in the non-traditional group. Furthermore, a clear gradient for sharing a household with the partner and feeling lonely, which can be seen as proxies of social support, was observed in the subgroups two to four, which might indicate as well that social support is an important gender-related factor. However, things looked somewhat different with view on hypertension: differences in prevalence remained almost the same and statistically significant within as well as across all the comparisons between men and women constituting the male-dominated and female-dominated subgroups. This observation might indicate that an intersectional perspective on gender as defined by the covariables of the IG-Score might not be as meaningful for hypertension as assumed with respect to the other health indicators. In this case it seems plausible that differences in hypertension might be driven more strongly by sex-related factors.

Sixth, to adopt the IG-Score approach as a standard practice in PHMR, it may be best to identify a specific set of intersectional variables that are most appropriate for each country. The possibilities of operationalizing the intersectionality of sex/gender depend primarily on the historical and cultural development of a country and thus also on the availability of corresponding data in order to be able to depict these aspects. So, for example, scholars who advocate for the implementation of intersectionality in quantitative health research recommend expanding the scope of intersectionality in quantitative health research from sex/gender and race/ethnicity to further domains of social position such as socioeconomic status, educational attainment, and age [1,3]. Although the application of the intersectionality framework in quantitative research is now well established beyond the categorical boundaries of sex/gender and race/ethnicity, some intersectionality researchers worry that this may cause the concept of intersectionality to lose its roots in Black feminism [1]. One possible reason for the different treatment especially of the category “race/ethnicity” in intersectionality-informed quantitative research could be that the history of the recording and availability of data about “race/ethnicity” is very different in many countries, e.g., in the European context [35]. The German state, with its National Socialist past, for example, has not collected any specific data on “race” and/or “ethnicity” since the end of World War II. Even in Germany’s micro-census, the largest representative survey conducted annually by the Federal Statistical Office since 1957, only limited data on residents’ country of origin are available [36]. Accordingly, the “origin” of the population is recorded in the micro-census primarily through information on citizenship [36]. In PHMR in Germany, the problem that data on the health situation of people with a migration background is very incomplete is publicly discussed [37]. Basically, people with a migration background in Germany are a very heterogeneous group, as they differ in aspects such as countries of birth, reasons for migration and length of stay in Germany [37,38]. In addition, the limitations that arise when describing population health in PHMR primarily along the category of citizenship are known [38]. However, even if information on the different countries of birth of the SOEP cohort is available, albeit only for a limited part of the study sample, the very heterogeneous composition of the immigrant population in Germany and the problem that the origin of persons with an indirect migration background cannot always be determined [39] rather argue against considering countries of birth as an attempt to approximate “race/ethnicity” when calculating an IG score in the context of German PHMR. On the other hand, a typical and widely used approach to capture sociocultural facets such as acculturation, which in a broader sense can be seen as an approximation to the construct “race/ethnicity” in population-based studies in Germany, is the migration background [36]. We were also able to adequately account for this in our analyses. The data available in the SOEP not only allow for a categorization by place of birth Germany (no migration background vs. direct migration background), but also take into account, in the case of individuals born in Germany, whether their parents (mother or father) have a migration background (if so, participants were classified as having an indirect migration background). In addition, as a further approximation of acculturation, we were able to consider the extent to which German is generally spoken at home. In summary, we were reluctant to attempt to operationalize “race” and/or “ethnicity” beyond this, as a non-discriminatory categorization of aspects related to “race” and/or “ethnicity” is a sensitive topic in Germany due to German history and continues to be the subject of political and scholarly debates [36,40]. In conclusion, because the complex and gendered facets of social experiences and the availability of data about different dimensions of sociocultural, sociodemographic, and socioeconomic variables may differ across countries and cultures, PHMR in different countries may need to rely on different sets of intersectional variables that best reflect country-specific intersectional feminization processes. Thus, these specific IG scores may capture the prevailing feminization processes in different countries, and trends in men’s and women’s health may be monitored and reported in light of the intersectional profiles over time.

Seventh, the conducted analysis relies on estimates of the IG score. Thus, a suitable classifier has to be chosen. GBM is a standard technique in machine learning and we have shown that it outperformed the classical approach of logistic regression in terms of AUC-ROC. Moreover, it does not rely on a-priori model specification as the latter is able to uncover complex covariable interactions. An additional advantage of this method is that it can utilize surrogate splits to handle missing values. This reduces the risk of introducing bias by running a complete case analysis or using an imputed dataset. Since the number of subgroups is another crucial parameter that should not be determined a-priori, we compared results of four to seven subgroups. This led to a selection of five clusters minimizing covariable imbalance.

Finally, some methodological limitations of the proposed strategy should be addressed as well. Although the cited R packages allow for an easy calculation of the IG-Score and assessment of balance, it can be exhaustive to explore several numbers of subgroups in finding the best balance. In addition to the number of subgroups finally selected, stratification might be sensitive to the chosen intersectional variables which are used to calculate the IG-Score. Nonetheless, we observed that stratification is mostly influenced through the variables with greatest imbalance in the full sample, i.e., employment status and occupational status. Furthermore, computing the IG-Score showed that roundabout 0.5% of the study sample had to be excluded due to no overlap in the distribution of the covariables between men and women used to compute the IG-Score. As PHMR aims to represent the general population, the excluded group may need further analysis to reflect and understand how gender and related roles are evolving across the entire population. For example, the group of women who had no IG-Score overlap with men of the study sample (Appendix A) was mainly characterized by living with children under 16 in the household (98%), being on maternity leave (98%), aged 18–39 years (90%), with no partner in the household (74%), and by direct or indirect migration background (46%). This intersectional profile at the group level could be of particular interest for planning PHMR focus reports, as these women are exposed to social experiences which do not seem to be reflected in the social experiences of men at the population level. Presumably, most women in this subgroup may not be able to participate in the paid labour force because they have primary responsibility for raising children and for the most part no partner with whom they can split both - breadwinning as well as homemaking responsibilities.

The presented strategy for population health analysis aimed to ensure the simultaneous consideration of sex (biological/physiological factors), gender (sociocultural factors, power relations), and their interrelationship with other intersecting social aspects such as age, immigration status, and socioeconomic status. This analytical perspective is consistent with the World Health Organization’s commitment to Sex- and Gender-Based Analysis (SGBA+) [41,42]. The potential of stratification on an Intersectional Gender-Score as an analytical approach to strengthen intersectionality-informed Public Health Monitoring and Reporting and its sex/gender sensitivity should be further explored.

## 5. Conclusions

Stratification on an IG-Score is particularly effective when Public Health Monitoring and Reporting is designed to avoid stigmatization based on sex/gender group assignment-especially when differences in the prevalence of, for example, mental health are observed. If differences in the prevalence of health indicators are observed in the overall sample but IG-Score-subgroups can be identified in which these differences are not present, this may suggest that these intersectional profiles can explain the sex/gender differences observed in the overall sample.

## Figures and Tables

**Table 1 ijerph-20-02220-t001:** Characterisation of male-dominated and female-dominated subgroup based on most strongly diverging intersectional variables between the subgroups.

Intersectional Variables	Male-Dominated Subgroup N = 4645 [Men: 82.41% Women: 17.59%]	Female-Dominated Subgroup N = 4632 [Men: 6.95% Women: 93.05%]
Employment status		
full-time part-time occasionally maternity/parental leave housewife/-husband Occupational status blue-collar white-collar else freelance	90.97% 0.40% 1.03% 0% 0% 35.57% 40.53% 4.79% 12.15%	0.36% 64.18% 11.56% 6.00% 16.30% 7.45% 62.79% 23.77% 1.31%

**Table 2 ijerph-20-02220-t002:** Prevalence of frequent mental distress (FMD), depression, self-rated health and hypertension within the total sample, the male-dominated and the female-dominated subgroup stratified by men/women.

	Full Sample		Male-Dominated Subgroup		Female-Dominated Subgroup	
N total	23,159		4645		4632	
	Men	Women	Men	Women	Men	Women
Sex % (n)	46.31 (10,724)	53.69 (12,435)	82.41 (3828)	17.59 (817)	6.95 (322)	93.05 (4310)
N	10654	12366	3806	816	322	4305
FMD (very)often, % (n) [CI]	6.93 (738) [6.45, 7.43]	14.55 (1799) [13.93, 15.18]	5.41 (206) [4.72, 6.18]	12.50 (102) [10.31, 14.97]	9.32 (30) [6.37, 13.03]	13.45 (579) [12.44, 14.51]
N	10724	12435	3828	817	322	4310
Depression yes, % (n) [CI]	5.17 (554) [4.75, 5.60]	9.61 (1195) [9.10, 10.14]	3.47 (133) [2.92, 4.10]	7.22 (59) [5.54, 9.22]	8.07 (26) [5.34, 11.61]	8.45 (364) [7.63, 9.32]
N	10714	12421	3825	815	322	4309
Self-rated health not good, % (n) [CI]	14.54 (1558) [13.88, 15.22]	17.91 (2224) [17.23, 18.59]	12.05 (461) [11.04, 13.12]	14.36 (117) [12.02, 16.95]	14.91 (48) [11.20, 19.27]	14.48 (624) [13.44, 15.57]
N	10724	12435	3828	817	322	4310
Hypertension yes, % (n) [CI]	24.20 (2595) [23.39, 25.02]	20.44 (2542) [19.74, 21.16]	20.90 (800) [19.62, 22.22]	14.93 (122) [12.56, 17.56]	21.43 (69) [17.07, 26.32]	15.01 (647) [13.96, 16.11]

Abbreviations: SD-standard deviation; CI-95% confidence interval.

**Table 3 ijerph-20-02220-t003:** Differences between men and women in prevalence of frequent mental distress (FMD), depression, self-rated health, and hypertension within and across the male-dominated subgroup and the female-dominated subgroup.

	FMD Difference of Prevalence in %	Depression Difference of Prevalence in %	Self-Rated Health Difference of Prevalence in %	Hypertension Difference of Prevalence in %
Differences between men and women				
Full sample	−7.62	−4.44	−3.37	3.76
[CI]	[−8.42, −6.83]	[−5.12, −3.77]	[−4.32, −2.41]	[2.67, 4.84]
Traditional gender S1 men-S5 women	−8.04	−4.98	−2.43	5.89
[CI]	[−9.31, −6.77]	[−6.01, −3.93]	[−3.93, −0.93]	[4.19, 7.58]
Male-dominated * S1 men-S1 women	−7.09	−3.75	−2.31	5.97
[CI]	[−9.54, −4.63]	[−5.69, −1.81]	[−5.00, 0.39]	[3.13, 8.80]
Female-dominated * S5 men-S5 women	−4.13	−0.38	0.43	6.42
[CI]	[−7.63, −0.63]	[−3.63, −2.89]	[−3.77, 4.62]	[1.64, 11.12]
Non-traditional gender S5 men-S1 women	−3.18	0.85	0.55	6.50
[CI]	[−7.30, 0.94]	[−2.83, 4.53]	[−4.24, 5.34]	[1.17, 11.82]
Differences between women				
S1 women-S5 women	−0.95	−1.23	−0.12	−0.08
[CI]	[−3.51, 1.61]	[−3.26, 0.81]	[−2.83, 2.57]	[−2.82, 2.66]
Differences between men				
S1 men-S5 men	−3.91	−4.60	−2.86	−0.53
[CI]	[−7.33, −0.48]	[−7.80, −1.40]	[−7.05, 1.34]	[−5.36, 4.30]

* Men and women are balanced with respect to sociocultural, sociodemographic, and socioeconomic characteristics. Abbreviations: S1-subgroup 1, S5-subgroup 5.

## Data Availability

Anonymized SOEP data are available as public use files only for research purposes (doi:10.5684/soep.v34).

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
