# Peer review of "Intersectionality-Informed Sex/Gender-Sensitivity in Public Health Monitoring and Reporting (PHMR): A Case Study Assessing Stratification on an “Intersectional Gender-Score”"

_ijerph, 2023, doi:10.3390/ijerph20032220_

Round 1

Reviewer 1 Report (Previous Reviewer 1)

Nicely written and nice revisions. This is an interesting topic which will add to the literature of intersectionality and health.

Reviewer 2 Report (Previous Reviewer 2)

The authors have addressed my comments and I am happy with the revision.

This manuscript is a resubmission of an earlier submission. The following is a list of the peer review reports and author responses from that submission.

Round 1

Reviewer 1 Report

This article discusses a more precise way to report disease prevalence and self-rated health in non-binary people using an intersectional gender score. Within the male-dominate IG score group, gender differences in disease prevalence remained. In the female dominated IG group, prevalence of disease was observed to be higher for men/women with higher female dominate IG scores.

Overall, it is very well written and an interesting article, which may change the way we think about male/female/non-binary people disease prevalence.

Abstract:

Provides a well written summary of the article.

Introduction:

This is very well written and easy to understand. Nice references. 

Line 48-50: the way it is written is confusing. Consider changing to “AdvanceDataAnalysis is part of the collaborative research network AdvanceGender, which aims to promote sex/gender-sensitive and intersectional health research and health reporting [4,5]. Using AdvanceDataAnalysis, we focus on the translation of social theory into quantitative methodology.”

Line 132: What survey? From the SOEP? Please specify.

Very clear aim for the manuscript listed in lines 81-89 but then it is restated and more specific in lines 138-147. You could consider combining the two sections for better flow.

Methods:

Line 183: the term “housewife/-husband- does this imply a wife or husband who does not work? Or do you just mean wife/husband? If you mean the former, wouldn’t this overlap with the “unemployed group”. Also, unemployed is listed twice.

Line 187: disability status: is this self-reported, medical diagnosis, or a legal assignment?

Line 194/197: Binarized implies “two” but you have more than 2 options for particpants to choose in each of these responses. Please change.

Results:

For table 1, it is very clearly presented. However, were are there any associated p values or CI to signify statistical significance?

Author Response

Response to Reviewer 1 Comments

Dear Reviewer,

Thank you for your thoughtful comments which have helped us to improve and strengthen our manuscript. Please find below our response to your comments (the numbering of the lines refers to the revised version with track changes):

Comments and Suggestions for Authors

This article discusses a more precise way to report disease prevalence and self-rated health in non-binary people using an intersectional gender score. Within the male-dominate IG score group, gender differences in disease prevalence remained. In the female dominated IG group, prevalence of disease was observed to be higher for men/women with higher female dominate IG scores.

Overall, it is very well written and an interesting article, which may change the way we think about male/female/non-binary people disease prevalence.

Abstract:

Provides a well written summary of the article.

Introduction:

This is very well written and easy to understand. Nice references. 

COMMENT: Line 48-50: the way it is written is confusing. Consider changing to “AdvanceDataAnalysis is part of the collaborative research network AdvanceGender, which aims to promote sex/gender-sensitive and intersectional health research and health reporting [4,5]. Using AdvanceDataAnalysis, we focus on the translation of social theory into quantitative methodology.”

RESPONSE: Thank you very much for pointing this out. Changes were made according to your recommendation. (lines 52-54)

COMMENT: Line 132: What survey? From the SOEP? Please specify.

RESPONSE: Yes, this is correct. Data are based on The German Socio-Economic Panel’ (SOEP). This is now specified in the manuscript. (line 152)

COMMENT: Very clear aim for the manuscript listed in lines 81-89 but then it is restated and more specific in lines 138-147. You could consider combining the two sections for better flow.

RESPONSE: We followed your recommendation and combined the two sections. (lines158-173)

Methods:

COMMENT: Line 183: the term “housewife/-husband- does this imply a wife or husband who does not work? Or do you just mean wife/husband? If you mean the former, wouldn’t this overlap with the “unemployed group”. Also, unemployed is listed twice.

RESPONSE: Thank you for this comment, as we missed to describe this category in more detail. We now specify the description of “unemployed” to “registered as unemployed” and “housewife/-husband” to “housewife/-husband (self-identified; caring for children, relatives and/or doing housework)” in the manuscript (lines 219-220)

COMMENT: Line 187: disability status: is this self-reported, medical diagnosis, or a legal assignment?

RESPONSE: Yes, it is a self-report of a legal assignment. This is now specified in the manuscript. (line 224)

COMMENT: Line 194/197: Binarized implies “two” but you have more than 2 options for particpants to choose in each of these responses. Please change.

RESPONSE: We made the changes accordingly and now describe the binarization of the variables in more detail (“How often have you felt sad?” [very often, often, occasionally, rarely, very rarely; binarized for analysis into very often/often vs occasionally/rarely/very rarely]” and “Third, self-rated health was assessed by the following question: “How would you describe your current health?” [bad, poor, satisfactory, good, very good; binarized for analysis into bad/poor vs satisfactory/good/very good ].” (lines 232-237)

Results:

COMMENT: For table 1, it is very clearly presented. However, were are there any associated p values or CI to signify statistical significance?

RESPONSE: We aim to present/characterize the two extreme groups (female- and male-dominated group) only descriptively.

Reviewer 2 Report

I was excited by the title and the methods are certainly robust.  There are a lot of results to read through and I think the main findings need clarifying for the reader as they get a bit lost in the text. Perhaps a summary of the main highlights could form the first paragraph of the discussion. Overall I feel this is an important addition to the evidence base on intersectionality and health but it could be improved in terms of the interpretation and reaching a wider audience. For example, what do the results mean in lay person language?

Author Response

Response to Reviewer 1 Comments

Dear Reviewer,

Thank you for your thoughtful comments which have helped us to improve and strengthen our manuscript. Please find below our response to your comments (the numbering of the lines refers to the revised version with track changes):

REVIEWER 2

Intersectionality-informed sex/gender-sensitivity in Public Health Monitoring and Reporting (PHMR): A case study assessing stratification on an “Intersectional Gender-Score”

Introduction

COMMENT: Line 42: Do you mean ‘multiply burden’? or do you mean ‘multiple burden’?

RESPONSE: This expression is a quote from Kimberlé Crenshaw's 1989 essay entitled "Demarginalising the Intersection of Race and Sex: A Black Feminist Critique of Antidiscrimination Doctrine, Feminist Theory and Antiracist Politics". We have added the inverted commas accordingly.

COMMENT: Line 54: ‘which might as well be adequate’ – suggest delete ‘as’

RESPONSE: Thank you for pointing this out, we deleted ‘as’ according to your suggestion.

COMMENT: Line 55: ‘Therefore’ rather than ‘Thereby’??

RESPONSE: Thank you for pointing this out, we changed ‘Therefore’ to ‘Thereby’ according to your suggestion.

COMMENT: Lines 92-108; lines 132-156 – does this not belong in the methods section?

RESPONSE: Thank you for the comment, which we can well understand. Since the manuscript was submitted in a methodologically oriented special issue and the other reviewer did not make any reference to the placement of the paragraph, we hope that you will agree that we leave the paragraph in the background. By placing the paragraph in the background, we want to sufficiently introduce the readership to our proposed methodological strategy before describing the statistical methods in detail.

COMMENT: Lines 149-156 are definitely too detailed for the introduction section.

RESPONSE: Thank you for pointing this out. We have shortened the paragraph considerably. (lines 173-176)

Materials and Methods

COMMENT: Line 191: Why were these health indicators chosen?? Detail is missing about the validation of these research questions. Is the limitation of self-report also discussed later in the paper?

RESPONSE: The health indicators were chosen from an intersectionality-informed perspective. Self-rated health, depression, frequent mental distress, and hypertension have already been associated with experiences of discrimination in a meta-analytic review and are often reported as different between men and women (lines 64-71and 163-166)

COMMENT: Line 248 delete repeated use of ‘it’

RESPONSE: The repeated use of ‘it’ has been deleted.

Results:

COMMENT: Line 272: what does ‘often missing company’ mean?

RESPONSE: Thank you very much for pointing this out. We changed the wording throughout the manuscript to “often feeling lonely”.

COMMENT: Line 284: ‘..served: In contrast..’ – grammatical error

RESPONSE: Changes have been made accordingly.

COMMENT: Lines 283-294 – has ‘S2-S4’ been defined/explained previously?

RESPONSE: ‘S2-S4’ is an abbreviation for subgroups 2-4. It has been defined previously in the manuscript (lines 375-376)

COMMENT: Tables 2 and 3 require definition of FMD

RESPONSE: Thank you for pointing this out. Changes have been made accordingly.

COMMENT: Lines 326 to 335 – please check grammar, particularly use of colons and full-stops.

RESPONSE: Changes have been made accordingly.

Discussion:

COMMENT: Line 371: ‘..first time were: In order..’ again please do not use a capital letter following a colon. Same again for line 487, line 541

RESPONSE: Changes have been made accordingly.

COMMENT: Line 392 – please end with a full stop here not a colon.

RESPONSE: Changes have been made accordingly.

COMMENT: Lines 367-392 – I don’t think this is necessary as it is simply a summary of the methods.

RESPONSE: Thank you for pointing this out. The whole paragraph has been deleted.

COMMENT: To me, lines 459 to 471 and 476 to 478, 483 to 486 are the most interesting – can this be reflected more in the abstract?

RESPONSE: Thank you very much for this comment. As we did not want to exceed the permissible word count of the manuscript, we added only one sentence to the summary.

COMMENT: Line 475: ‘..IG-Score: In CTA..’ again please do not use a capital letter following a colon.

RESPONSE: Changes have been made accordingly.

COMMENT: Line 540 – suggest changing to ‘..the potential applications and limitations..’??

RESPONSE: Thank you for the suggestion. Changes have been made accordingly.

COMMENT: Lines 539-545: I think the conclusion needs further work, it needs to be more succinct, perhaps move some text into the discussion section.

RESPONSE: Following your suggestion, the conclusion is now more succinct and a whole paragraph has been moved to the discussion.

COMMENT: Last sentence 559 to 561 – move to discussion as I would note end on this note.

RESPONSE: Thank you very much for pointing this out to us. We have also moved the sentence to the discussion section.